# Semi-supervised Source Detection in Astronomical Images: New Benchmark and Strong Baseline

## Abstract

Source detection in modern observational astronomy is a cornerstone for accurately localizing and identifying stellar sources. It is crucial for studies such as stellar population synthesis and cosmological parameter estimation. However, the characteristics of astronomical images, including high density, the effect of point spread functions and low signal-to-noise ratios, significantly challenge the latest advanced object detectors. Besides, fully-supervised detection methods are hardly practical, due to the significant difficulty in annotating dense, small, and faint sources in astronomical images. To tackle the scarcity of astronomical datasets, we introduce a new comprehensive benchmark (LAMOST-DET), comprising 18,400 astronomical images and 728,898 source instances. Upon the dataset, we further devise a novel semi-supervised learning framework coined Nova Teacher, capable of detecting dense sources effectively given sparse annotations. It integrates source light enhancement module, confidence-guided pseudo-supervision, and cross-view complementary mining in a dual-teacher paradigm. Extensive experiments on LAMOST-DET show that, Nova Teacher consistently improves previous competitors by 4.04% and 5.22% mAP under two semi-supervised settings. Additionally, our method competes against other detectors on a natural image dataset, validating its generalization ability to various scenarios. Source codes and data samples are available at supple. material.

## 1 Introduction

In the domain of modern observational astronomy, source detection has long been a cornerstone for precise localization and identification of stellar sources, and underpins extensive studies ranging from stellar population synthesis and galactic structure mapping to cosmological parameter estimation and astrometric measurements Lang et al. (2016); Xu & Zhu (2024). In general, each astronomical source is detected as an ellipse using a five-parameter representation, including the center coordinates $(x_c, y_c)$, the major semi-axis $a$, the minor semi-axis $b$, and the orientation angle $\theta$, as shown in Fig. 1. Despite its centrality, the task of source detection is highly challenged by several astronomy-specific factors, including high density, the effect of point spread functions (PSF), and low signal-to-noise ratio (SNR) Savage & Oliver (2007); Bertin & Arnouts (1996). For instance, Figure 1 shows the astronomical image observed by Large Sky Area Multi-Object Fiber Spectroscopic Telescope (LAMOST) Su et al. (1998). Given these complexities, the recall rate for source detection often suffers a significant decline.

To address these challenges, traditional source detection methods Mouhcine et al. (2005); Zheng et al. (2015); Hausen & Robertson (2020) typically employ several image processing techniques including but not limited to photometric measurements Stetson (1987), threshold detection Zheng et al. (2015) and watershed algorithms Hausen & Robertson (2020). These techniques perform adequately in simple imaging scenarios, whereas they primarily rely on low-level and hand-crafted features, hindering their adaptability to complex and dynamic scenarios heavily. Moreover, each image observed by astronomical surveys contains thousands of sources, making these computationally expensive traditional methods hardly applicable to large-scale astronomical survey data.

Figure 1: Illustration of source detection in astronomical images, including (i) the LAMOST imaging system; (ii) high-resolution (HR) image of astronomical sources; (iii) challenges leading to recall degradation, including: (a) high source density, (b) PSF effects, and (c) low SNR. Each source is normally detected as an ellipse object based on five-parameter regression.

On the other hand, recent advancements in deep learning have introduced a new perspective for source detection Burke et al. (2019); Van Oort et al. (2019); Karmakar et al. (2018). These methods offer more powerful feature extraction, adapt well to diverse imaging conditions, and significantly improve detection efficiency. Although these high-performance detectors are capable of handling large-scale source detection, they still rely on fully annotated datasets available during training. However, astronomical sources are often small and densely packed, and require more expert knowledge for accurate annotation. Hence, a fully-supervised learning paradigm is nearly impractical to astronomical source detection any more. For instance, the LAMOST imaging system can observe up to 1200 light sources in a single observation, but we can note that it is infeasible to annotate all the sources completely. Instead, semi-supervised object detection (SSOD) methods Chen et al. (2021; 2022); Jeong et al. (2019); Li et al. (2020; 2024) have gained significant attention due to their ability to effectively detect objects with limited annotations. Nevertheless, such methods still assume the availability of a substantial amount of fully-labeled images, and tend to struggle in astronomical domains with extremely sparse annotations.

To this end, we propose a new benchmark and label-efficient approach for semi-supervised source detection. First of all, to tackle the scarcity of astronomical image datasets, we construct a new benchmark (**LAMOST-DET**) based on the LAMOST imaging system, containing 4,600 high-resolution images, each of which is further separated into four quadrants, resulting in a set of 18,400 images with calibrated astrometric and photometric information. Building upon this dataset, we propose a novel semi-supervised learning framework coined **Nova Teacher**, leveraging a dual-teacher paradigm and pseudo-label guidance to enhance recall rates and effectively detect dense astronomical sources in the context of sparse annotations. Concretely, our method introduces key innovations in three aspects. 1) We devise a source light enhancement module (SLEM) capable of enhancing weak source signals while suppressing complex background, leading to improved detection of small and dense targets. 2) We design a confidence-guided pseudo-supervision (CGPS) mechanism, strategically exploiting confidence scores to stratify pseudo-labels, as a result of high-confidence pseudo-labels acting as positive supervision, and low-confidence ones being repurposed to guide negative sample mining. 3) We further develop a cross-view complementary mining (CVCM) strategy to enrich the pool of pseudo-label candidates, thereby promoting the discovery of unlabeled and hard-to-detect light sources. Thanks to the effectiveness of the components above, our method not only achieves state-of-the-art performance on the LAMOST-DET benchmark, but also competes with previous approaches on a widely used dataset of natural images.

The main contributions are summarized as follows:

- We introduce LAMOST-DET, a comprehensive benchmark comprising 18,400 astronomical images observed by LAMOST and 728,898 astronomical source instances, providing a powerful platform for the challenging task of semi-supervised source detection.

- We propose a novel semi-supervised framework coined Nova Teacher, which combines source light enhancement module, confidence-guided pseudo-supervision, and cross-view complementary mining in a dual-teacher paradigm, to achieve dense source detection under the constraints imposed by annotation scarcity.

- Our method outperforms other semi-supervised object detectors on the LAMOST-DET dataset, where we achieve significant gains of 4.04% and 5.22% mAP on two data split settings. Besides, our Nova Teacher competes with previous approaches on a natural image dataset, signifying its strong generalization ability to various scenarios.

## 2 RELATED WORK

**Astronomical Source Detection.** Traditional source detection methods in astronomy primarily rely on classical image processing techniques, such as adaptive thresholding (*e.g.* , OTSU Zheng et al. (2015), SExtractor Bertin & Arnouts (1996)), differential photometry Stetson (1987), watershed algorithms Hausen & Robertson (2020), and edge detectors Mouhcine et al. (2005). While effective for images with low noise and simple backgrounds, such methods depend on low-level hand-crafted features, but lack adaptability to complex astronomical scenes. Besides, they are hindered by complicated processes and slow computation, limiting their scalability for large-scale astronomical data. With the advent of deep learning, a new generation of methods has achieved remarkable success in source detection, for example the Mask R-CNN based framework for source detection in panoramic images Burke et al. (2019); Long et al. (2015), U-Net based solutions for precise source localization Richards et al. (2023); Van Oort et al. (2019), and a VAE-GMM hybrid model for stellar identification Karmakar et al. (2018). These methods demonstrate impressive accuracy and flexibility, and excel in regions with complex morphological structures. However, most existing detectors assume that all astronomical sources are fully annotated for supervised learning, which is not practical for astronomical images that often contain thousands of sources simultaneously (Fig. 1). Unlike natural images, annotating astronomical sources is not only costly but also ambiguous. Particularly when dealing with faint or crowded sources, it is heavily reliant on expert knowledge. To overcome the limitations, *this work is the first to explore a semi-supervised learning framework for accurate and efficient source detection without relying on extensive manual annotations.*

**Semi-supervised Object Detection.** Typically, SSOD is built upon a teacher-student paradigm, where the teacher model generates pseudo-labels for unlabeled data to guide the student model Chen et al. (2021; 2022); Jeong et al. (2019); Li et al. (2020); Wang et al. (2021b). While extensive advances (*e.g.* , Unbiased Teacher Liu et al. (2021), Soft Teacher Xu et al. (2021), Dense Teacher Zhou et al. (2022), Consistent Teacher Wang et al. (2023b), and Focal Teacher Wang et al. (2024)) have introduced new mechanisms to improve pseudo-label quality or consistency, their methods still assume the availability of a substantial amount of fully labeled data, and are hardly applicable in scenarios with extremely limited annotations. Sparsely annotated object detection (SAOD) Zhang et al. (2020); Yoon et al. (2021), which is a special case of semi-supervised object detection (SSOD), has gained increasing attention as its ability for leveraging sparsely labeled instances Zhang et al. (2020); Yoon et al. (2021). For instance, Siamese network Wang et al. (2021a), dual-stream network Suri et al. (2023) and calibration mechanism Wang et al. (2023a) are developed to further improve the utilization of sparsely labeled images. Nevertheless, most of existing methods are primarily designed for natural images with rectangular bounding boxes, but cannot generalize well to the elliptical sources which are highly clustered in astronomical images. *This motivates our proposed approach, which is specifically designed to address the unique difficulties raised by source detection in astronomical images.*

## 3 LAMOST-DET BENCHMARK

The scarcity of publicly available astronomical image datasets has significantly hindered progress in source detection. This section introduces LAMOST-DET, a new and comprehensive dataset curated specifically to tackle the lack of datasets in source detection.

**Observational Data.** The data used in this study were observed with the Large Sky Area Multi-Object Fiber Spectroscopic Telescope (LAMOST) Su et al. (1998), also known as the Guo Shoujing Telescope. The dataset includes both imaging and spectroscopic observations across galactic and extragalactic fields, covering dense stellar regions as well as sparse areas. Sky coverage, field selection, and exposure times adhere to the LAMOST survey strategy, allowing thousands of sources to be observed simultaneously in each image. We capture observational data of 4,600 high-resolution images, which are preprocessed following a comprehensive scheme, including bias calibration, flat-

field correction, fiber tracking, sky subtraction, wavelength calibration, exposure merging and wave-length band connection. After filtering and removing redundant data from the same set of exposures, the available signals from four quadrants are saved separately to form the final dataset, consisting of 18,400 images with accurately calibrated astrometric and photometric information.

**Ground-truth Annotations.** Labeling the sources involves both automatic and manual procedures. First, source candidates in each image are identified automatically by the Source Extraction and Photometry (SEP) library, a Python implementation of SExtractor Bertin & Arnouts (1996). Each image undergoes median filtering and background subtraction to enhance the detection reliability. After automatic detection, spurious sources are removed through manual reviews conducted by astronomical experts. Note that, the original image $I$ should be converted to gray-scale level using the *z-scale* normalization for visual analysis:

$$I' = \text{clip}\left(\frac{I - z_1}{z_2 - z_1} \times 255, \ 0, \ 255\right), \tag{1}$$

where $z_1$ and $z_2$ are determined by the *z-scale* algorithm, and $I'$ is the converted image.

**Data Statistics.** We obtain 728,898 annotated astronomical sources from 18,400 images. The images are divided into train (60%), validation (20%), and test (20%) sets, respectively. As shown in Fig. 2, we shed more light on the data distributions: (a) we first count the number of astronomical sources in each image and then separate the images into different groups based on their source numbers, for train/val/test subsets. (b) we measure the detection difficulty by signal-to-noise ratio (SNR) and group the astronomical sources into three levels, extreme $(0, 5]$, moderate $(5, 15]$, and trivial $(15, \infty]$. These data statistics highlight the challenges in terms of spatial density and detection difficulty. We can expect that, LAMOST-DET serves as a potential and strong foundation for benchmarking detection models under various conditions.

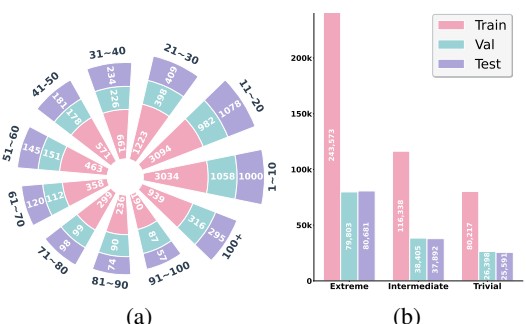

(a)            (b)

Figure 2: (a) Data statistics according to the number of astronomical sources per image across train (pink), val (green), and test (purple) subsets. (b) Dividing the astronomical sources into three groups (Extreme, Intermediate and Trivial) by measuring detection difficulty based on SNR.

In particular, we define two protocols in the experiments to make this dataset tailored for semi-supervised source detection. *Additional details are elaborated in supple. material.*

## 4 METHODOLOGY

Source detection is to identify elliptical sources using a five-parameter regression representation. To address semi-supervised source detection, we propose a novel **Nova Teacher** framework, which is mainly composed of source light enhancement module, confidence-guided pseudo-supervision mechanism, and cross-view complementary mining strategy, as depicted in Fig. 3.

### 4.1 SOURCE LIGHT ENHANCEMENT MODULE (SLEM)

Detecting faint astronomical sources under sparse annotations is particularly challenging due to the low signal-to-noise ratio (SNR) and the effects of the point spread function (PSF), which often cause the sources to appear as blurred spots, hindering accurate localization and identification. To tackle it, we propose a source light enhancement module, which selectively amplifies weak astronomical source signals while suppressing complex background. The whole SLEM integrates several key steps as follows.

First of all, *non-linear intensity transformation* is applied to highlight faint source signals while adaptively controlling the enhancement of bright regions. The transformation is expressed as:

$$F_{enh} = \text{sign}(I) \cdot |I| \cdot [1 - \exp(-\alpha \cdot s \cdot |I|)], \tag{2}$$

where $I$ is the input image, $\alpha$ is a fixed parameter and $s$ is a dynamic scaling factor determined by the mean intensity.

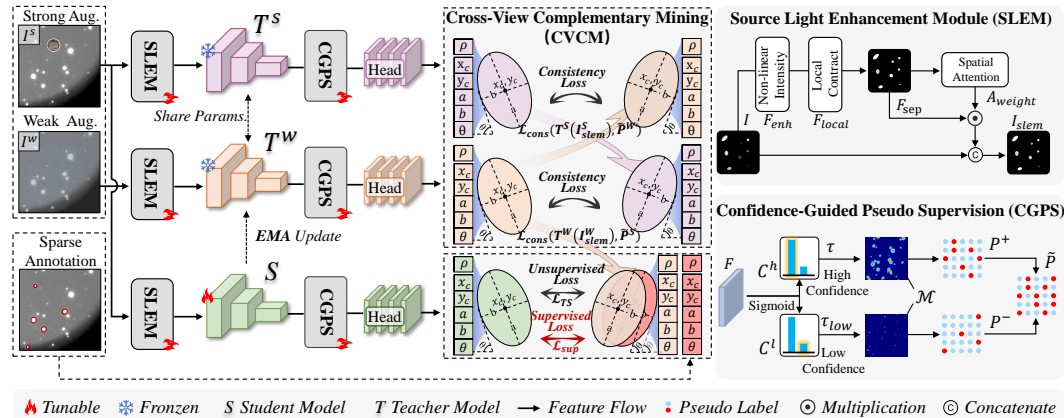

Figure 3: **Overview framework of our Nova Teacher for semi-supervised source detection.** The source light enhancement module (SLEM) enhances faint source features while suppressing complex backgrounds. Then, two teacher models, $T^S$ and $T^W$, generate pseudo-labels from weak and strong augmentation views, respectively. These pseudo-labels are further refined using the confidence-guided pseudo-supervision (CGPS) mechanism, which divides regions into high- and low-confidence areas. At last, cross-view complementary mining (CVCM) leverages complementary pseudo-labels from both teachers to guide the student model $S$.

To further enhance the separation of sources from the background, we employ *local contrast enhancement*, so that high-intensity regions are suppressed, whereas faint sources can be accentuated. The process begins with $\gamma = 1 + \tanh(-2 \cdot F_{mean})$, where $F_{mean}$ is the local mean intensity computed by applying a $5 \times 5$ average pooling to $F_{enh}$. As a result, we obtain the locally enhanced feature $F_{local} = F_{enh} \cdot \gamma$. Then, $F_{local}$ is processed using depth-wise separable convolution to extract features efficiently:

$$F_{sep} = \sigma(\text{Conv}_{\text{point}}(\sigma(\text{Conv}_{\text{depth}}(F_{local})))), \tag{3}$$

where $\text{Conv}_{\text{depth}}$ and $\text{Conv}_{\text{point}}$ are depth-wise and point-wise convolutions, and $\sigma$ represents the leaky ReLU activation. Afterwards, we introduce spatial attention to make the detector focus more on the enhanced regions. The spatial attention weights $A_{weight}$ are computed as follows:

$$A_{weight} = \text{Sigmoid}(\text{Conv}(\text{Concat}[A_{avg}; A_{max}])), \tag{4}$$

where $A_{avg}$ is global average pooling over $F_{sep}$, and $A_{max}$ is global max pooling over $F_{sep}$.

The final output is obtained by concatenating the original input image with the attention-weighted enhanced features, followed by a $1 \times 1$ convolution:

$$I_{slem} = \text{Conv}_{1 \times 1}(\text{Concat}[I; F_{sep} \odot A_{weight}]), \tag{5}$$

where $\odot$ represents element-wise multiplication. $I_{slem}$ is the resulting image enhanced by SLEM, and is passed on to subsequent detection modules in Nova Teacher.

## 4.2 CONFIDENCE-GUIDED PSEUDO-SUPERVISION (CGPS)

In astronomical images, the dense packing of sources in high-density regions makes individual identification challenging, while PSF further blurs these sources, and under low SNR conditions, weak astronomical signals are often overwhelmed by noise, leading to both false positives and missed detections. These challenges often lead to uncertainty in the pseudo-labels during training, and their errors may harm model performance. Existing pseudo-labeling methods overlook the differences in label reliability and treat negative samples as noise. However, negative samples may contain important information that distinguishes the astronomical sources. Unlike prior work, we design a confidence-guided pseudo-supervision (CGPS) mechanism between the backbone network and the detection head, which effectively leverages negative samples to improve the quality of pseudo-labels.

First of all, given the feature representation $F$ extracted from the enhanced image $I_{slem}$, we generate reliable confidence matrix with $C = \text{Sigmoid}(\text{Conv}_{1 \times 1}(F)) \in \mathbb{R}^{H \times W}$. Next, we assign pseudo-labels to high-confidence and low-confidence groups based on the confidence scores:

$$C^h = \{i \in C \mid \tilde{c}_i > \tau\}, \ C^l = \{i \in C \mid \tilde{c}_i < \tau_{low}\}, \tag{6}$$

where $\tilde{c}_i$ represents the confidence score of pseudo-label $i$; $\tau$ and $\tau_{low}$ are empirical thresholds for high- and low-confidence pseudo-labels. We note that, high-confidence pseudo-labels $C^h$ are considered reliable and are prioritized as positive supervision for training. Meanwhile, low-confidence pseudo-labels $C^l$ contain potential extra information, especially about the difficult-to-detect sources, and are treated as hard-negative samples to improve model discriminability.

Subsequently, we construct a pseudo-label heatmap $M \in \mathbb{R}^{H \times W}$ based on these confidence scores, reflecting label information for each position:

$$\mathcal{M}_{x,y} = \begin{cases} \tilde{c}_i, & \text{if } (x,y) \in bbox_i \text{ and } i \in C^h \\ -\tilde{c}_j, & \text{if } (x,y) \in bbox_j \text{ and } j \in C^l, \end{cases} \quad (7)$$

where $bbox_i$ and $bbox_j$ represent the bounding boxes of the $i$-th and $j$-th pseudo-labels in the high-confidence and low-confidence regions, respectively. Here, each pixel in $\mathcal{M}$ is assigned with a weight based on its position: for high-confidence areas, the region is set with the positive confidence score $\tilde{c}_i$, while low-confidence areas are with the negative confidence score $-\tilde{c}_j$ to adjust the impact of negative samples. Ultimately, CGPS uses high- and low-confidence regions in the heatmap $\mathcal{M}$ for positive pseudo-labels $P^+$ and negative pseudo-labels $P^-$, respectively, and combines $P^+$ and $P^-$ to obtain the complete pseudo-labels $\tilde{P}$.

## 4.3 CROSS-VIEW COMPLEMENTARY MINING (CVCM)

Due to label scarcity, existing pseudo-labeling-based semi-supervised methods struggle to effectively focus on the unlabeled regions, resulting in a training bottleneck where recall is hard to improve. To unlock the potential of pseudo-labels, we propose a cross-view complementary mining (CVCM) strategy, which leverages complementary information from different perspectives to uncover "unseen" sources and enrich the pseudo-label candidate pool.

To be specific, we adopt two views with different levels of augmentation: a weakly augmented view and a strongly augmented view, whose complementarity improves detection recall. To maximize the utilization of these pseudo-labels, CVCM employs a complementary mining strategy: the weak augmentation teacher $T^W$ generates preliminary pseudo-labels $\tilde{y}^W$ to guide the strong augmentation teacher $T^S$, while $T^S$ generates more challenging pseudo-labels $\tilde{y}^S$ to guide $T^W$ in return. This complementary supervision allows the model to identify and mine "unseen" sources that may be overlooked in a single view, thereby expanding the pseudo-label set. Finally, we formulate CVCM by integrating two consistency loss functions:

$$\mathcal{L}_{cvcm} = \frac{1}{|\tilde{P}^W|} \sum_{i \in \tilde{P}^W} \mathcal{L}_{cons}\left(T^S(I^S_{slem})_i, \tilde{P}^W_i\right) + \frac{1}{|\tilde{P}^S|} \sum_{j \in \tilde{P}^W} \mathcal{L}_{cons}\left(T^W(I^W_{slem})_j, \tilde{P}^S_j\right),$$
$$(8)$$

where $I^W_{slem}$ and $I^S_{slem}$ are the images enhanced by SLEM; $\tilde{P}^W$ and $\tilde{P}^S$ denote the selected pseudo-label regions. The consistency loss $\mathcal{L}_{cons}$ includes classification loss $\mathcal{L}_{cls}$ and regression loss $\mathcal{L}_{reg}$. Here, we define $\mathcal{L}_{cls}$ with binary cross-entropy cost, and $\mathcal{L}_{reg}$ as $L_1$ cost.

## 4.4 OVERALL LOSS FUNCTION

The objective of optimizing Nova Teacher integrates supervised learning with sparsely labeled data and semi-supervised learning with pseudo-labeled data:

$$\mathcal{L}_{total} = \mathcal{L}_{sup} + \omega_u \mathcal{L}_{unsup} = \mathcal{L}_{sup} + \omega_u(\lambda \mathcal{L}_{cvcm} + \mathcal{L}_{ts}), \quad (9)$$

where $\mathcal{L}_{sup}$ corresponds to the supervised loss which we define in the *supple. material* due to limited space. $\omega_u$ is the unsupervised loss weight, which we set to 2 following prior work. $\lambda$ adjusts the impact of CVCM. The unsupervised loss $\mathcal{L}_{unsup}$ combines $\mathcal{L}_{cvcm}$ and $\mathcal{L}_{ts}$. $\mathcal{L}_{ts}$ is computed between the teacher $T^W$ and the student $S$:

$$\mathcal{L}_{ts} = \sum_{i=1}^{N} \mu_i^{rot}\left(\mathcal{L}_{cls}(\widehat{p}_i, \tilde{p}_i) + \mathcal{L}_{reg}(\widehat{\delta}_i, \tilde{\delta}_i) + \mathcal{L}_{ctr}(\widehat{s}_i, \tilde{s}_i)\right), \quad (10)$$

where $\mathcal{L}_{cls}$ is binary cross-entropy loss, $\mathcal{L}_{reg}$ is $L_1$ loss, and $\mathcal{L}_{ctr}$ is centrality score loss; $\widehat{p}_i, \widehat{\delta}_i, \widehat{s}_i$ are student predictions, while $\tilde{p}_i, \tilde{\delta}_i, \tilde{s}_i$ are pseudo-labels from the weak teacher; $\mu_i^{rot}$ is a spatial weighting factor.

Table 1: Quantitative comparisons on the LAMOST test set. 50%, 70% and 90% of the training data are unlabeled, based on two different data splits (Split-1 and Split-2), respectively. mRecall and mAP represent the mean recall and mean average precision.

| Method | Split-1 | | | | | | Split-2 | | | | | |
| | 50% | | 70% | | 90% | | 50% | | 70% | | 90% | |
| | mRecall | mAP | mRecall | mAP | mRecall | mAP | mRecall | mAP | mRecall | mAP | mRecall | mAP |
|---|---|---|---|---|---|---|---|---|---|---|---|---|
| Faster R-CNN Girshick (2015) | 38.78 | 35.36 | 36.32 | 33.46 | 28.41 | 26.78 | 33.02 | 34.99 | 39.53 | 34.91 | 23.97 | 25.77 |
| Oriented R-CNN Xie et al. (2021) | 40.73 | 42.88 | 38.56 | 35.17 | 27.01 | 26.52 | 36.12 | 35.51 | 42.87 | 42.97 | 42.01 | 42.41 |
| Rotated FCOS Tian et al. (2019) | 69.31 | 54.52 | 68.86 | 54.06 | 67.53 | 52.74 | 67.34 | 54.16 | 63.86 | 53.77 | 65.87 | 52.55 |
| Rotated RetinaNet Lin et al. (2017b) | 65.91 | 53.67 | 65.55 | 53.56 | 63.60 | 51.21 | 64.65 | 52.57 | 64.51 | 52.73 | 63.39 | 51.86 |
| Co-mining Wang et al. (2021a) | 67.46 | 53.79 | 67.01 | 54.39 | 66.83 | 53.07 | 67.71 | 54.28 | 67.02 | 54.14 | 66.95 | 53.38 |
| Dense Teacher Zhou et al. (2022) | 68.67 | 55.51 | 68.32 | 54.43 | 66.62 | 54.11 | 69.87 | 55.09 | _69.64_ | 54.56 | 67.86 | 53.26 |
| SOOD Hua et al. (2023) | _68.75_ | 55.69 | _68.94_ | 54.66 | _68.12_ | 53.68 | _69.91_ | 55.14 | 69.51 | 54.57 | _68.31_ | 53.73 |
| Focal Teacher Wang et al. (2024) | 66.82 | _57.24_ | 68.29 | _55.25_ | 66.31 | _54.38_ | 68.26 | _56.08_ | 67.95 | _55.36_ | 66.83 | _54.45_ |
| **Nova Teacher (Ours)** | **70.84** | **59.64** | **70.38** | **58.09** | **68.48** | **55.83** | **71.01** | **59.01** | **70.34** | **58.21** | **69.79** | **57.33** |

## 5 EXPERIMENT

**Dataset Protocols.** Recall that LAMOST-DET contains 728,898 astronomical sources existing with 18,400 images. Following the settings in current benchmarks (*e.g.* , PASCAL VOC and MSCOCO), we design two different protocols splitting LAMOST-DET suitable for semi-supervised source detection. **(1) Split-1:** for each image in the training set, we randomly remove $p\%$ of the annotations in this image. This image-level manner results in all training images having incomplete annotations. We experiment with three sparsity levels where $p = \{50, 70, 90\}$. **(2) Split-2:** we randomly remove $p\%$ of the annotations belonging to all the images of the training set. This partitioning strategy, referred to as overall sparsity, may lead to some images containing no annotations at all. Likewise, we set $p = \{50, 70, 90\}$. In terms of evaluation performance, we adhere to two commonly-used quantitative metrics including mean recall (**mRecall**) and mean average precision (**mAP**). The threshold for determining true positives is set to 0.5.

**Implementation Details.** Without loss of generality, our model is built upon FCOS Tian et al. (2019) as it is a representative anchor-free detector. We utilize ResNet-50 He et al. (2016) with FPN Lin et al. (2017a) as the backbone. Inspired by prior work Tarvainen & Valpola (2017); Liu et al. (2021); Zhou et al. (2022), we adopt a "burn-in" strategy for initializing the two teacher networks in Nova Teacher. The model is trained for 120k iterations on a single NVIDIA RTX 4090 GPU. We use a SGD optimizer with an initial learning rate of 0.0025, which is reduced by a factor of 10 at the 80k and 110k iterations. Momentum and weight decay are set to 0.9 and 0.0001. The exponential moving average (EMA) momentum for both teachers is set to 0.9996.

### 5.1 MAIN RESULTS

This section reports the overall comparison under two data split settings. Since there have no existing methods specifically designed for semi-supervised source detection, we thereby re-implement several widely-benchmarked object detectors using the sparse annotations of LAMOST-DET, and compare their performance with our Nova Teacher for fair evaluations. The compared baselines include Faster R-CNN Girshick (2015), Oriented R-CNN Xie et al. (2021), Rotated FCOS Tian et al. (2019), and Rotated RetinaNet Lin et al. (2017b), as well as four state-of-the-art semi-supervised object detectors: Co-mining Wang et al. (2021a), Dense Teacher Zhou et al. (2022), SOOD Hua et al. (2023), and Focal Teacher Wang et al. (2024). Note that, one primary reason for selecting these baselines is because they can be successfully adjusted from bounding-box regression to ellipse regression with minimal cost. Below, we present both quantitative and qualitative results.

**Quantitative Comparison.** Table 1 summarizes the performance of all compared methods on the LAMOST-DET test set. We can see that, Nova Teacher achieves the highest mRecall and mAP scores for both Split-1 and Split-2 settings, covering three sparsity levels (*i.e.* 50%, 70% and 90%). For the Split-1 setting, Nova Teacher achieves a large performance gain over several popular methods like Rotated FCOS, and consistently outperforms recent strong baselines such as SOOD and Focal Teacher. For instance, at a sparsity level of 50%, Nova Teacher achieves an mRecall of 71.01% and an mAP of 59.01%, which are 6.02% and 4.19% higher than the recent popular method Focal Teacher, respectively. In terms of the Split-2 setting, we similarly obtain the best results across all three sparsity levels. For example, at 90% sparsity level, mRecall and mAP reach 69.79% and 57.33%, respectively, showing an improvement of at least 4.43% and 5.29% over the other methods.

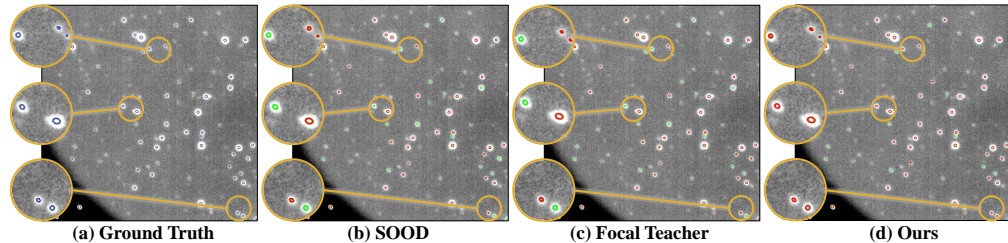

| (a) Ground Truth | (b) SOOD | (c) Focal Teacher | (d) Ours |

Figure 4: Qualitative comparison with other semi-supervised methods. The blue markers represent the ground truth, the green markers represent undetected positive samples, and the red markers represent the predicted results.

These results demonstrate the superiority of Nova Teacher for semi-supervised source detection in context of limited annotations.

**Qualitative Comparison.** We further conduct a qualitative comparison between Nova Teacher and other competitors, as shown in Fig. 4. It can be seen that, Nova Teacher achieves higher recall and more precise boundary predictions compared to other methods. In the three zoomed-in regions, Dense Teacher, SOOD and Focal Teacher all fail to detect one source target. In contrast, Nova Teacher successfully detects that source, demonstrating superior source localization. Nevertheless, all methods encounter failures in detecting sources with very low signal-to-noise ratios, highlighting the difficulty in source detection.

**Recall Performance Across Different SNR Intervals.**

Recall is a critical metric as it measures the model's ability to correctly identify sources. Due to the influence of noise and variations in source characteristics, recall tends to be relatively low in source detection tasks. To assess recall performance, we evaluate it across different signal-to-noise ratio (SNR) intervals, with the results summarized in Table 2. The Nova Teacher method consistently outperforms all other approaches, achieving recall rates of 66.13%, 76.27%, and 77.21% in the Extreme, Intermediate, and Trivial SNR categories, respectively. These results represent improvements of 2.83%, 1.38%, and 0.91% over the second-best performer, SOOD.

Table 2: Recall rates of semi-supervised methods at different SNR intervals.

| Method | Extreme | Intermediate | Trivial |
|---|---|---|---|
| Dense Teacher | 63.39 | 74.74 | 75.80 |
| SOOD | 63.30 | 74.89 | 76.30 |
| Focal Teacher | 60.71 | 72.73 | 76.52 |
| Nova Teacher (ours) | 66.13 | 76.27 | 77.21 |

## 5.2 ABLATION STUDY AND ANALYSIS

We conduct ablation experiments to verify the effectiveness of our proposed components. Experiments are based on the Split-1 setting where 50% annotations are removed.

**Component Analysis.** Table 3 show the effectiveness of all components designed for Nova Teacher. Note that, M1 refers to the ablation model which is composed of one teacher and one student, whereas other models (M2-6) are built based on two teachers and one student. First, we can see that, M1 underperforms all the remaining models, validating the benefit of combining two teachers. In addition, these ablation results prove that each component in our model contributes to the performance gains.

Table 3: Ablation study results on LAMOST-DET test set.

| Model | SLEM | CVCM | CGPS | mRecall | mAP |
|---|---|---|---|---|---|
| M1 | ✗ | ✗ | ✗ | 68.75 | 55.69 |
| M2 | ✓ | ✗ | ✗ | 69.18 | 57.31 |
| M3 | ✗ | ✓ | ✗ | 70.26 | 56.03 |
| M4 | ✓ | ✓ | ✗ | 70.39 | 56.84 |
| M5 | ✗ | ✓ | ✓ | 70.28 | 58.07 |
| M6 | ✓ | ✓ | ✓ | **70.84** | **59.64** |

**Impact of Loss Weight $\lambda$.** This experiment investigates the effect of the loss weight $\lambda$ (refer to Eq. 9) balancing the teacher-student loss $\mathcal{L}_{ts}$ and cross-view complementary mining loss $\mathcal{L}_{cvcm}$. Figure 5 presents the performance when we vary $\lambda$ from 0.2 to 1.0. The best results, with an mRecall score of 70.84% and an mAP score of 59.64%, are achieved when $\lambda$ is set to 0.8. Decreasing $\lambda$ to 0.2 leads to

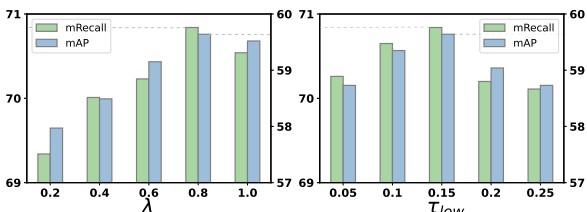

Figure 5: Impact of the loss weight $\lambda$ and confidence threshold $\tau_{low}$ on our model.

a substantial performance decline, primarily due to its eliminating correct pseudo-labels obtained by

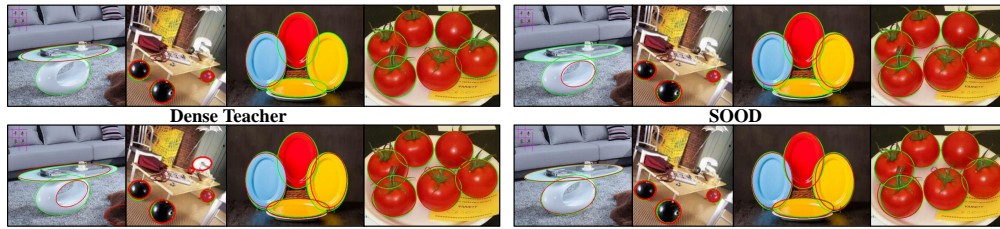

Figure 6: Qualitative comparison with other semi-supervised methods on General Ellipse Detector benchmark. Green marker represents true values, red marker indicates predicted results.

CVCM. Conversely, increasing $\lambda$ may excessively introduce noisy pseudo-labels, thereby harming the overall accuracy largely.

**Impact of Confidence Threshold** $\tau_{low}$. This experiment aims to study the impact of the low-confidence threshold $\tau_{low}$ used in the CGPS module for selecting hard negative pseudo-labels. We note that, increasing $\tau_{low}$ makes more pseudo-labels be hard negatives, enhancing the detector's robustness to difficult cases. However, setting $\tau_{low}$ too high tends to introduce more noisy or incorrect pseudo-labels, which may degrade the performance significantly. As shown in Fig. 5, our results indicate that when $\tau_{low} = 0.15$, it strikes the best balance between these effects, and achieves the optimal overall performance.

### 5.3 EVALUATION ON GENERAL ELLIPSE DETECTION

Although Nova Teacher is designed for astronomical source detection, it should be potentially applicable to natural image datasets. To prove it, we conduct additional experiments on the General Ellipse Detector (GED) benchmark Wang et al. (2022), where we compares Nova Teacher against other mainstream detectors. We evaluate the methods under three sparsity levels of the Split-2 setting. As shown in Table 4, Nova Teacher consistently outperforms all other methods. Particularly, regarding the 90% sparsity level, our mRecall and mAP arrive at 66.72% and 41.22%, respectively, showing significant gains over other methods. We can

Table 4: Performance comparison on the GED benchmark using the Split-2 setting.

| Method | 50% | | 70% | | 90% | |
|---|---|---|---|---|---|---|
| | mRecall | mAP | mRecall | mAP | mRecall | mAP |
| Faster R-CNN | 85.26 | 77.85 | 77.87 | 65.73 | 34.92 | 23.31 |
| Oriented R-CNN | 86.38 | 78.26 | 84.59 | 73.08 | 44.44 | 32.33 |
| Rotated FCOS | 84.96 | 77.24 | 80.82 | 70.77 | 33.54 | 23.69 |
| Rotated RetinaNet | 91.16 | 84.23 | 85.37 | 74.03 | 51.68 | 36.88 |
| Dense Teacher | 90.14 | 81.67 | 87.15 | 74.46 | 46.62 | 27.96 |
| SOOD | 91.30 | 83.82 | 87.29 | 74.36 | 50.25 | 29.17 |
| Focal Teacher | 90.53 | 84.36 | 88.79 | 75.67 | 62.20 | 36.49 |
| **Nova Teacher** | **92.39** | **85.87** | **89.11** | **76.73** | **66.72** | **41.22** |

expect the promising performance achieved by Nova Teacher, as general objects in those natural images are not so complex than the sources in astronomical images.

We further carry on a qualitative comparison, as shown in Fig. 6. Overall, our method provides more accurate boundaries and angles than other methods. Concretely, in the first column, Dense Teacher, SOOD, and Focal Teacher, produce inaccurate detection boxes. In the second column, Focal Teacher generates a false positive (*i.e.*, "white sign"). In the third column, several targets are missed by the other methods. Lastly, in the fourth column, all methods missed one of the ellipse targets, specifically the "tomato" ellipse.

## 6 CONCLUSION

In this paper, we have proposed a novel semi-supervised approach for source detection in astronomical images. To solve the scarcity of astronomical data, we have constructed LAMOST-DET, a comprehensive benchmark comprising 18,400 astronomical images and 728,898 astronomical source instances. Built upon this, we introduce a novel semi-supervised framework namely Nova Teacher, through integrating source light enhancement module, confidence-guided pseudo-supervision, and cross-view complementary mining, to accomplish dense source detection under sparse annotation constraints. Extensive experiments indicate that our method not only achieves state-of-the-art performance on LAMOST-DET, but also effectively competes with previous approaches on a natural image dataset. This work promotes research and applications in source detection, and we will focus more on leveraging multi-modal large models to further improve pseudo-label learning.

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

# A    APPENDIX

In this document, we elaborate more details on the LAMOST-DET benchmark, the Nova Teacher model and the compared experiments, respectively. In addition, source codes and some data samples are attached with this document as well. The whole LAMOST-DET dataset will be publicly available soon after the review period.

## A.1    MORE ON THE LAMOST-DET BENCHMARK

The Large Sky Area Multi-Object Fiber Spectroscopic Telescope (LAMOST), also known as the Guo Shoujing Telescope, is an advanced astronomical observatory designed to conduct large-scale spectroscopic surveys Su et al. (1998). Located at the Xinglong Station of the National Astronomical Observatories, Chinese Academy of Sciences, LAMOST features a unique optical design that combines a large aperture with a wide field of view, making it an essential tool for studying the universe.

### A.1.1    OPTICAL DESIGN AND WORKING PRINCIPLE

LAMOST employs a specialized reflecting Schmidt optical system, which is optimized for wide-field, multi-object spectroscopy, as illustrated in Fig. 7. Its primary mirror (Mb) measures 6.67 m by 6.05 m and is segmented into 37 hexagonal sub-mirrors. The active Schmidt mirror (Ma), which has dimensions of 5.74 m by 4.40 m, is composed of 24 hexagonal sub-mirrors that can adjust their shape in real-time, ensuring optimal optical performance during observations. This active optics system is crucial for correcting distortions caused by atmospheric conditions and the telescope's own structural deformations.

The focal surface of LAMOST is equipped with 4000 optical fibers, each positioned precisely to capture light from individual astronomical objects. These fibers feed the collected light into 16 spectrographs, which are equipped with CCD cameras to record the spectral data. The system covers wavelengths from 370 nm to 900 nm, with a spectral resolution ranging from $R = 1000$ to 5000 depending on the configuration. This allows LAMOST to conduct detailed spectroscopic surveys of millions of objects in the universe, including stars, galaxies, and quasars.

### A.1.2    OBSERVATIONAL PROCESS AND DATA ACQUISITION

LAMOST's automated and efficient observational process enables large-scale surveys, allowing it to observe vast sky areas in a single pointing and collect data on thousands of objects simultaneously through its unique focal plane and fiber positioning system.

The telescope's optical design, combined with an advanced fiber positioning system, enables it to perform simultaneous multi-object spectroscopy with high throughput. The system uses a parallel-controllable fiber positioning technique, allowing for precise and flexible placement of the fibers on the focal surface. This is particularly advantageous for observing regions with dense star fields or crowded extragalactic environments, where traditional single-object observation methods would be inefficient.

Once the light is collected by the fibers, it is directed to the spectrographs, which disperse the light into its component wavelengths. The spectrographs are equipped with volume-phase holographic (VPH) gratings, which allow for high-resolution spectral analysis across a wide wavelength range. The data collected by the spectrographs are then processed through a series of calibration steps, including bias subtraction, flat-field correction, and wavelength calibration, ensuring that the final spectral data are accurate and reliable.

### A.1.3    DATA PROCESSING AND PIPELINE

The raw data collected by LAMOST are processed through a comprehensive pipeline that includes several stages of calibration and analysis. The data preprocessing begins with bias subtraction to remove any electronic noise from the detectors, followed by flat-field correction to account for variations in the sensitivity of the detector pixels. Fiber tracking is then performed to ensure that the light from each astronomical object is correctly assigned to the corresponding fiber.

Figure 7: Diagram illustrating the optical and data flow systems of LAMOST. The system consists of an active Schmidt optical configuration, a fiber positioning system, and 16 spectrographs.

Wavelength calibration is a critical step, as it ensures that the spectral lines in the data are accurately mapped to their corresponding wavelengths. This is followed by sky subtraction to remove any contributions from the night sky, which can contaminate the measurements of celestial objects.

Once the raw data is calibrated, it is processed further to extract the one-dimensional spectra of the observed objects. The LAMOST pipeline is designed to handle large volumes of data efficiently, making it possible to conduct high-throughput surveys and generate large datasets for further analysis.

### A.1.4 DETAILS ON *zscale* NORMALIZATION

The *zscale* normalization is a widely used technique in astronomical image processing for adaptively mapping the pixel intensity range of a FITS image to a displayable grayscale interval, typically $[0, 255]$. This approach enhances the visibility of both faint and bright features, facilitating more effective use of the image by downstream neural network models.

The zscale algorithm operates as follows. First, a subset of pixel values is uniformly sampled from the input image to capture the overall intensity distribution. These sampled values are sorted, and the median intensity $m$ is computed. To estimate the scaling, a robust linear fit (often a least-squares regression with outlier rejection) is performed on the pixel values as a function of their rank order, yielding a slope parameter $s$. Using a contrast parameter $c$ (with a typical default value of $0.25$), the algorithm defines a mapping window $[z_1, z_2]$ centered on the median, calculated by:

$$z_1 = m - \frac{n}{2} \times s \times c, \quad z_2 = m + \frac{n}{2} \times s \times c, \tag{11}$$

where $n$ is the number of sampled pixels. Finally, apply Eq. 1 from the main manuscript to linearly map each pixel value $I$ in the image to the grayscale range $[0, 255]$. This adaptive normalization ensures that both faint and bright astronomical sources are well represented in the final PNG image, while remaining robust to outliers and extreme values.

### A.2 MORE ON METHODOLOGY

### A.2.1 MORE PRELIMINARIES ON ELLIPSE REPRESENTATION

In source detection tasks, sources are typically modeled as elliptical regions, parameterized by five values: $(x_c, y_c, a, b, \theta)$. Here, $(x_c, y_c)$ denote the center coordinates, $a$ and $b$ are the semi-major

and semi-minor axes reflecting the spatial extent of the source, and $\theta \in [-\frac{\pi}{2}, \frac{\pi}{2})$ specifies the orientation angle. This regression-based framework enables precise localization and morphological characterization of sources, which is crucial for resolving densely packed or overlapping sources in astronomical images.

We employ a single-stage, anchor-free detection architecture Tian et al. (2019), utilizing ResNet-50 He et al. (2016) as the backbone and a feature pyramid network (FPN) Lin et al. (2017a) for multi-scale feature extraction. The backbone produces hierarchical feature maps, which the FPN refines to yield feature representations $F_i \in \mathbb{R}^{H \times W \times 256}$, supporting subsequent regression and classification tasks.

At each spatial location $(x, y)$ on the feature map $F_i$, the detection head predicts: (1) a probability score $p_{x,y}$ representing the likelihood of a source; (2) a centrality score $s_{x,y}$ indicating proximity to the source center; (3) a set of regression offsets $(\Delta l, \Delta t, \Delta r, \Delta b, \theta)$, which denote the distances from $(x, y)$ to the source's boundary along the left, top, right, and bottom directions, as well as the orientation angle, as illustrated in Fig. 8. These parameters can be decoded to the five-parameter representation as follows:

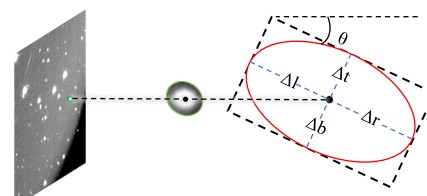

Figure 8: Regression of elliptical parameters for astronomical source detection.

$$\begin{cases} x_c = x + \cos(\theta) \cdot \dfrac{\Delta r - \Delta l}{2} - \sin(\theta) \cdot \dfrac{\Delta b - \Delta t}{2}, \\[2mm] y_c = y + \sin(\theta) \cdot \dfrac{\Delta r - \Delta l}{2} + \cos(\theta) \cdot \dfrac{\Delta b - \Delta t}{2}, \\[2mm] a = \dfrac{\Delta r + \Delta l}{2}, \qquad b = \dfrac{\Delta t + \Delta b}{2}, \qquad \theta = \theta. \end{cases} \tag{12}$$

Source detection is treated as a binary classification problem, with astronomical source regions as the foreground class and all other pixels as background. For each predicted probability $p_{x,y}$, the class with the highest probability is chosen as the final result. A spatial position $(x, y)$ is assigned as a positive sample if it falls within the ground-truth boundary of a source; otherwise, it is regarded as background.

For training, the ground truth consists of the five source parameters $\delta^* = (x_c, y_c, a, b, \theta)$ and the centrality score $s$. The centrality score $s^*$ is calculated as:

$$s_{x,y}^* = \sqrt{\frac{\min(\Delta l^*, \Delta r^*)}{\max(\Delta l^*, \Delta r^*)} \times \frac{\min(\Delta t^*, \Delta b^*)}{\max(\Delta t^*, \Delta b^*)}}, \tag{13}$$

where $(\Delta l^, \Delta t^, \Delta r^, \Delta b^)$ are the distances from $(x, y)$ to the true source boundaries. The centrality term encourages the model to focus on high-quality detections near the center of sources while suppressing low-quality bounding boxes. The loss function for training the source detection model combines classification loss, regression loss, and center-point loss:

$$\mathcal{L}_{sup} = \frac{1}{N_{pos}} \sum_{x,y} \mathcal{L}_{cls}(p_{x,y}, y_{x,y}^*) + \frac{1}{N_{pos}} \sum_{x,y} \mathcal{L}_{reg}(\delta_{x,y}, t_{x,y}^*) + \frac{1}{N_{pos}} \sum_{x,y} \mathcal{L}_{ctr}(s_{x,y}, s_{x,y}^*), \tag{14}$$

where $\mathcal{L}_{cls}$ is the focal loss for classification, $\mathcal{L}_{reg}$ is the rotated IoU loss for parameter regression, and $\mathcal{L}_{ctr}$ is the binary cross-entropy loss for centrality. $N_{pos}$ denotes the number of positive samples, and $*$ indicates the corresponding ground-truth values.

### A.2.2 MORE ON ALGORITHM PROCEDURE

In addition to the details provided in the main text, the training process of the Nova Teacher framework is outlined in Algorithm 1. This algorithm utilizes a teacher-student framework to optimize the astronomical image source detection model via semi-supervised learning with sparse annotations.

---

**Algorithm 1** Nova Teacher Training Procedure

---

1: **Input:** Raw astronomical images $I_{raw}$, Sparse ground-truth annotations $\mathcal{GT}_{sparse}$, Current number of iterations $k_{current}$
2: **Parameters:** $k_{burn\_in}$ (Number of burn-in iterations), $\omega_u$ (Hyperparameter for unsupervised loss), $\lambda$ (Hyperparameter for loss balance), $k_{total}$ (Total number of iterations for training)
3: **Output:** Updated student model $S$, Updated teacher model $T^W$
4:   # Training update strategy for each iteration
5: **if** $k_{current} > k_{burn\_in}$    and    $k_{current} \leq k_{total}$ **then**
6:     # Supervised loss with labeled samples
7:     $\mathcal{L}_{sup} \leftarrow$ supervised_loss$(P_S, \mathcal{GT}_{sparse})$, (Eq. 14)
8:     # Augmentation of raw image samples
9:     $I^W, I^S \leftarrow$ augment$(I_{raw})$
10:     # Source source enhancement via the SLEM module
11:     $I^S_{slem} \leftarrow$ SLEM$(I^W)$,  $I^S_{slem} \leftarrow$ SLEM$(I^S)$
12:     # Generate pseudo-labels and student predictions
13:     $\tilde{P}^W \leftarrow T^W(I^W_{slem})$,  $\tilde{P}^S \leftarrow T^S(I^S_{slem})$,  $\widehat{P} \leftarrow S(I^S_{slem})$
14:     # Compute consistency loss between $T^W$ and $T^S$
15:     $\mathcal{L}_{cvcm} \leftarrow \mathcal{L}_{cons}(T^S(I^S_{slem}), \tilde{P}^W) + \mathcal{L}_{cons}(T^W(I^W_{slem}), \tilde{P}^S)$, (Eq. 8)
16:     # Compute unsupervised loss between $S$ and $T^W$
17:     $\mathcal{L}_{ts} \leftarrow$ unsupervised_loss$(\widehat{P}, \tilde{P}^W)$, (Eq. 10)
18:     # Calculate the total loss cost
19:     $\mathcal{L}_{total} \leftarrow \mathcal{L}_{sup} + \omega_u(\mathcal{L}_{ts} + \lambda\mathcal{L}_{cvcm})$, (Eq. 9)
20:     # Update the teacher model via EMA
21:     update_teacher$(T^W, S)$,  share_parameters$(T^S, T^W)$
22: **end if**

---

Algorithm 1 presents the complete training procedure for the Nova Teacher framework, which employs a teacher-student architecture for semi-supervised astronomical image source detection with sparse annotations. The algorithm initializes with raw astronomical images $I_{raw}$ and sparse annotations $\mathcal{GT}_{sparse}$, along with key hyperparameters: burn-in iterations $k_{burn\_in}$, unsupervised loss weight $\omega_u$, loss balance parameter $\lambda$, and total iterations $k_{total}$. Each training iteration proceeds as follows: First, the supervised loss $\mathcal{L}_{sup}$ is computed using sparse labels. Data augmentation generates weakly and strongly augmented samples $I^W$ and $I^S$, which are subsequently enhanced through the SLEM module to produce $I^W_{slem}$ and $I^S_{slem}$. The teacher models $T^W$ and $T^S$ generate pseudo-labels $\tilde{P}^W$ and $\tilde{P}^S$ using $I^S_{slem}$ and $I^W_{slem}$, respectively, while the student model $S$ predicts $\widehat{P}$ for $I^S_{slem}$. The framework enforces consistency through two mechanisms: the consistency loss $\mathcal{L}_{cvcm}$ between teacher model outputs, and the unsupervised loss $\mathcal{L}_{ts}$ measuring discrepancy between student predictions and teacher pseudo-labels. The total loss $\mathcal{L}_{total}$ combines all components with their respective weights. Finally, the teacher model $T^W$ undergoes exponential moving average (EMA) updates, with parameters shared to the student model $T^S$.

## A.3 MORE ON EXPERIMENTAL DETAILS

### A.3.1 MORE ON IMPLEMENTATION DETAILS

Without loss of generality, our model is built upon FCOS Tian et al. (2019) as it is a representative anchor-free detector. We utilize ResNet-50 He et al. (2016) with FPN Lin et al. (2017a) as the backbone. In addition, asymmetric data augmentation is applied, including weak augmentation via random flipping Zhang et al. (2022); Li et al. (2022), and strong augmentation through random flipping, color jitter, random grayscale, and random Gaussian blur Dong et al. (2018); Tang et al. (2021). Inspired by prior work Tarvainen & Valpola (2017); Liu et al. (2021); Zhou et al. (2022), we adopt a "burn-in" strategy for initializing the two teacher networks in Nova Teacher. The model was trained for 120k iterations on a single NVIDIA RTX 4090 GPU. We use a SGD optimizer with an initial learning rate of 0.0025, which is reduced by a factor of 10 at the 80k and 110k iterations. Momentum and weight decay are set to 0.9 and 0.0001. Additionally, the exponential moving average (EMA) momentum for the teacher networks is set to 0.9996.

### A.3.2 Evaluation Metrics

To evaluate the performance of the proposed method, we use two widely adopted metrics in object detection: mean Recall (mRecall) and mean Average Precision (mAP).

**Mean Recall (mRecall)** measures the proportion of true positives correctly identified by the model relative to the total relevant objects in the ground truth. It is defined as:

$$\text{mRecall} = \frac{\sum_{i=1}^{N} \text{TP}_i}{\sum_{i=1}^{N} (\text{TP}_i + \text{FN}_i)}, \tag{15}$$

where $\text{TP}_i$ denotes the number of true positives, with the IoU threshold for determining TP set to 0.5, and $\text{FN}_i$ represents the false negatives for the $i$-th class or category. This metric reflects the model's ability to detect all relevant objects, considering both the recall of individual classes and the overall recall across all classes.

**Mean Average Precision (mAP)** is a widely used evaluation metric in object detection, which summarizes the precision-recall trade-off for each class. It calculates the average precision (AP) for each class, and the mAP is then the mean of the APs over all classes. The precision for a given class is defined as:

$$\text{Precision} = \frac{\text{TP}}{\text{TP} + \text{FP}}, \tag{16}$$

where TP denotes the number of true positives, and FP represents the false positives. The average precision (AP) for each class is computed by averaging the precision values over different recall levels. The mAP is obtained by averaging the APs of all classes in the detection task:

$$\text{mAP} = \frac{1}{C} \sum_{i=1}^{C} \text{AP}_i, \tag{17}$$

where $C$ is the total number of classes, and $\text{AP}_i$ is the average precision for class $i$. mAP provides a comprehensive measure of the model's accuracy across all classes, balancing both precision and recall.

### A.3.3 More Qualitative Results on LAMOST-DET

To complement the qualitative results presented in the main text, we conduct a comprehensive comparative analysis as shown in Figure 9. This evaluation compares performance across four baseline detectors—Faster R-CNN Girshick (2015), Oriented R-CNN Xie et al. (2021), Rotated RetinaNet Lin et al. (2017b), and Rotated FCOS Tian et al. (2019)—alongside three state-of-the-art semi-supervised methods: Dense Teacher Zhou et al. (2022), SOOD Hua et al. (2023), and Focal Teacher Wang et al. (2024), as well as our proposed Nova Teacher. The results demonstrate that Nova Teacher consistently outperforms both fully supervised approaches (*e.g.*, Faster R-CNN, Oriented R-CNN) and semi-supervised counterparts (*e.g.*, Dense Teacher, SOOD), achieving superior recall performance across diverse detection scenarios.

Specifically, in columns 1-2, Nova Teacher demonstrates significantly higher recall rates, successfully detecting the majority of small sources that other methods consistently miss, as evidenced by the reduced number of green annotations (undetected sources) compared to baseline methods like Faster R-CNN and Oriented R-CNN. Columns 3-4 further highlight this advantage, where Nova Teacher maintains robust detection capabilities in complex backgrounds, while Dense Teacher and SOOD exhibit substantial recall degradation with numerous missed detections marked in green. Most notably, in columns 5-6 featuring the most challenging scenarios with dense target distributions, Nova Teacher achieves near-complete target coverage, whereas competing methods including Focal Teacher show significant recall failures.

### A.3.4 More Qualitative Results on GED

As shown in Fig. 10, we further evaluate the qualitative comparison of Nova Teacher on GED with other methods, including Faster R-CNN Girshick (2015), Oriented R-CNN Xie et al. (2021), Rotated

FCOS Tian et al. (2019), Rotated RetinaNet Lin et al. (2017b), Dense Teacher Zhou et al. (2022), SOOD Hua et al. (2023), and Focal Teacher Wang et al. (2024). It can be seen that, Nova Teacher achieves higher recall and more precise boundary predictions compared to other methods.

In particular, while conventional methods such as Faster R-CNN, Rotated FCOS, Dense Teacher, and SOOD frequently fail to detect targets (*e.g.*, red plate in column 1, lollipops in column 2), Nova Teacher consistently identifies all objects with accurate bounding box regression. Moreover, when competing methods do achieve detection, they often suffer from imprecise localization (Oriented R-CNN, Rotated RetinaNet) or generate false positives (Focal Teacher's misclassification of white furniture in column 3, Oriented R-CNN's bicycle lock false alarm in column 5). Nova Teacher's robustness is particularly evident in challenging scenarios involving complex backgrounds and objects with large aspect ratios (column 4's coffee table detection), where it remains the sole method capable of accurate dual-target identification. Nevertheless, extremely sparse targets in cluttered environments (column 6's tomato) continue to pose detection challenges across all evaluated methods, indicating areas for future improvement in handling heavily occluded or background-integrated objects.

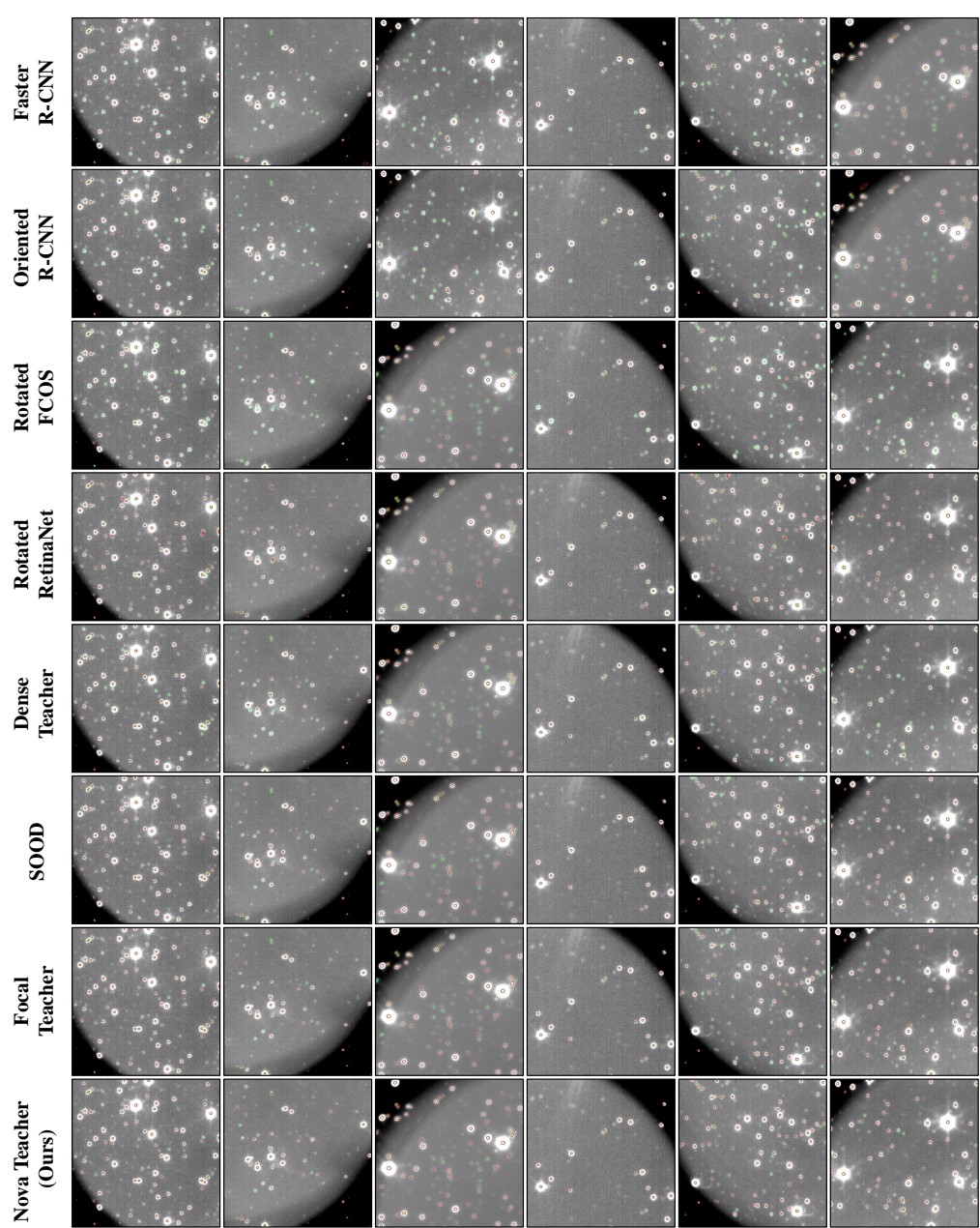

Figure 9: Qualitative comparison with other methods on the LAMOST dataset. The green marker represents the ground truth, while the red marker indicates the predicted values.

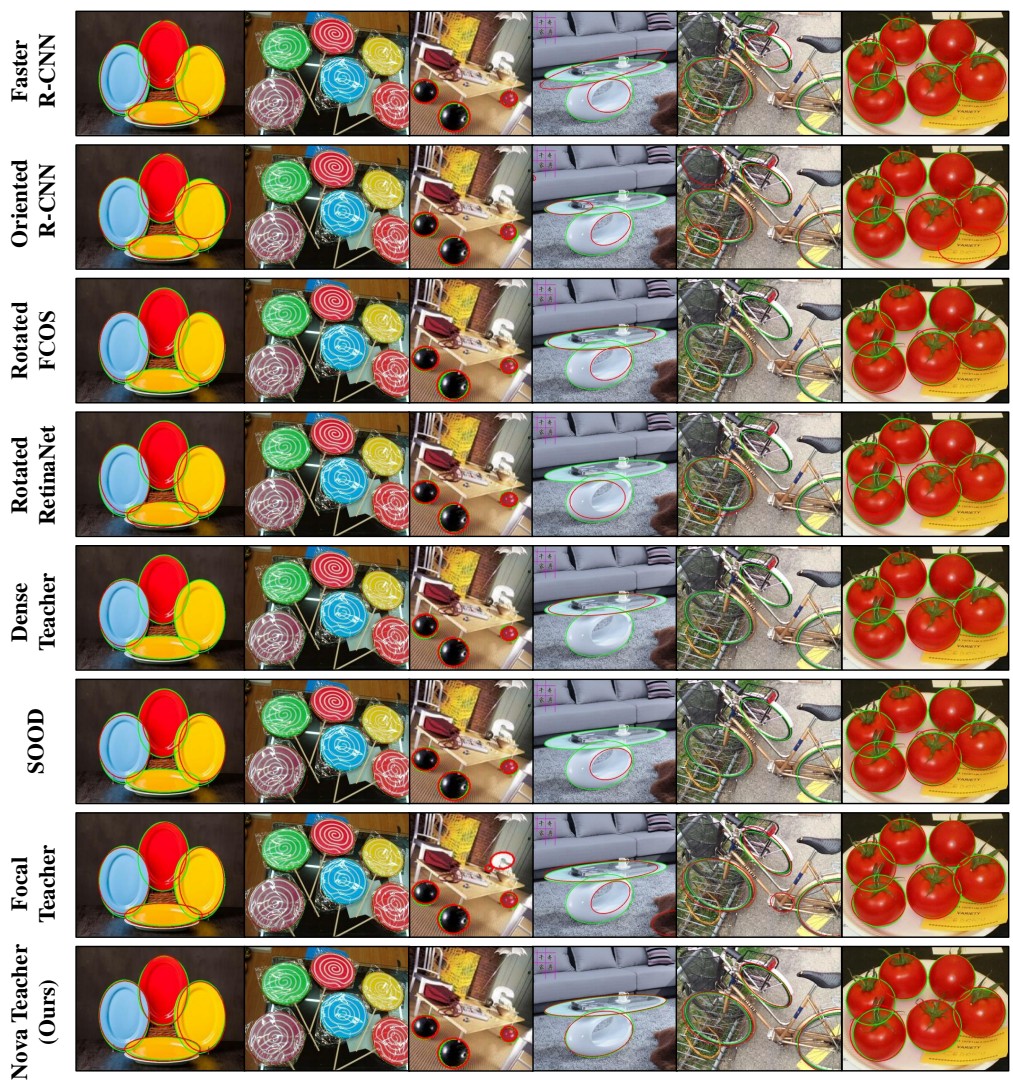

Figure 10: Qualitative comparison with other methods on the GED dataset. The green marker represents the ground truth, while the red marker indicates the predicted values.

