# OpenReview forum: "Semi-supervised Source Detection in Astronomical Images: New Benchmark and Strong Baseline"
_ICLR.cc/2026/Conference — ICLR 2026 Conference Withdrawn Submission_

### Official Review · Reviewer_RBKN · 2025-10-24

**Soundness:** 2
**Presentation:** 3
**Contribution:** 2
**Rating:** 4
**Confidence:** 3

**Summary:**

The paper focuses on semi-supervised source extraction for astronomy, proposing a new benchmark and method. The benchmark consists of cutouts of the LAMOST survey, annotated with SExtractor and further labeled with metadata on the number of sources and their S/N ratios. The method has a student-teacher set-up and comes with three modules, designed to enhance the signal in the complex images, deal with label noise, and enhance recall. The network is trained end-to-end with a combination of five losses and evaluated on the proposed benchmark against baselines from natural image (semi-)supervised object detection.

**Strengths:**

- The proposed setting is well-motivated.
- The figures enhance the clarity of the paper.
- The ablation studies show the benefit of the different components.

**Weaknesses:**

My main concerns lie in the experimental protocol.
First, the ground truth used for the evaluation of the methods is based on SExtractor, with an unknown number of spurious detections manually removed. However, SExtractor is known to struggle with fainter sources and achieves a much lower accuracy (albeit faster processing times) than many methods proposed since 1996 [1]. Using these detections as the ground truth thus optimizes for a sub-optimal ground truth. The analysis also does not contain a comparison to SExtractor itself or other unsupervised source extraction tools such as MTObjects or NoiseChisel. Given that SExtractor provides the ground truth, why not use SExtractor?
As a result, the results would be far more convincing if a simulated dataset were used for training and validation, where the exact ground truth is known, as also done in previous works [1]. This would allow for a comparison against standard methods and lead to more certainty in results. Additionally, it would allow for generalization experiments on real data, which would strengthen the paper, as currently the methods are only evaluated on a single astronomical dataset.
Lastly, splitting images into quadrants, then randomly dividing them into train and test, does not allow us to draw conclusions about generalization to less similar environments and backgrounds, which will be the reality in deployment.

Regarding the model, from the ablation in Table 3, it looks like the authors built on top of SOOD, as the ablation M1 results are the same as those for SOOD in Table 2. It would be appropriate to include how many additional parameters are added by the proposed modules to exclude the possibility that a bigger model is the reason for the performance increase.

[1] Haigh, Caroline, et al. "Optimising and comparing source-extraction tools using objective segmentation quality criteria." Astronomy & Astrophysics 645 (2021): A107.

**Questions:**

- How many spurious detections were "removed through manual reviews"?
- Why are the images split into quadrants? This way, quadrants from the same image will be present in both training and testing. To test generalization, it would be better to train on distinct cutouts from the test set.
- It is unclear why " We can expect the promising performance achieved by Nova Teacher, as general objects in those natural images are not so complex than the sources in astronomical images." (L465-467). The contributions in this paper are made specifically to tackle the difficulties present in astronomical images only. Why then should we expect better performance on natural images?
- How were the hyperparameters for the baselines set? Are they using the standard settings for natural images, or were they tuned for the task, as was done for the proposed model?

Other:
- Make sure to use \citep{} instead of \cite when a reference is not used as a part of the sentence itself.
- SSOD is defined a second time on L141
- "which we set to 2 following prior work" on L316 should contain a reference.
- How do you plan to use "multi-modal large models" (L486) for pseudo-labels? This is the first mention of multi-modality.

---

### Official Review · Reviewer_U26t · 2025-10-28

**Soundness:** 3
**Presentation:** 3
**Contribution:** 3
**Rating:** 6
**Confidence:** 3

**Summary:**

This paper proposes Nova Teacher, a semi-supervised framework for astronomical source detection under sparse annotations. It introduces a new benchmark, LAMOST-DET, containing over 18,000 images and 700k annotated sources. The framework integrates a source light enhancement module, confidence-guided pseudo-supervision, and cross-view complementary mining to improve recall and robustness. Overall, the writing is clear and well-organized, and the experiments are extensive. The contributions are technically solid, though some aspects, such as annotation quality and baseline novelty, could be further clarified.

**Strengths:**

1.	The writing is clear and easy to follow.
2.	The LAMOST-DET dataset is large-scale and contributes a valuable new resource.
3.	The proposed method is conceptually simple and achieves strong results.
4.	Experiments are thorough and convincingly demonstrate the method’s effectiveness.

**Weaknesses:**

1.	The paper lacks a detailed description of the annotation protocol; for example, the ellipses are not perfectly edge-aligned, raising concerns about label consistency.
2.	Comparisons with recent semi-supervised detection methods from 2025 are missing, which limits the assessment of competitiveness.
3.	The proposed CGPS and CVCM modules are conceptually similar to existing pseudo-labeling and mean teacher strategies; it would help to highlight their specific adaptation to astronomical source detection.

**Questions:**

1.	Could the authors elaborate on how annotation consistency was ensured given the imperfect alignment of elliptical masks?
2.	Will the dataset include detailed labeling guidelines or inter-annotator agreement statistics?
3.	How would Nova Teacher compare against the most recent 2025 semi-supervised detection frameworks?
4.	What are the distinctive aspects of CGPS and CVCM that make them particularly effective for elliptical source detection?

---

### Official Review · Reviewer_tvnt · 2025-10-31

**Soundness:** 2
**Presentation:** 2
**Contribution:** 2
**Rating:** 4
**Confidence:** 3

**Summary:**

This paper addresses the challenge of source detection in astronomical imaging by introducing a novel semi-supervised approach designed to enhance detection performance over existing methods. The proposed architecture consists of three key modules: a source light enhancement module, a confidence-guided pseudo supervision mechanism, and a cross-view complementary mining strategy. In addition to methodological innovations, the authors present a new benchmark dataset specifically for source detection in astronomical images. The effectiveness of the approach is demonstrated through comprehensive evaluations on this benchmark in a semi-supervised setting, where it achieves superior detection performance compared to previous methods. Furthermore, the method is tested on a non-astronomical task (semi-supervised ellipse detection in natural images) and again outperforms baseline approaches.

**Strengths:**

- The proposed benchmark is substantial in scale, encompassing a large number of images and annotated sources, which facilitates evaluation and comparison of models.

- The method demonstrates superior detection performance in both astronomical and natural image domains, indicating its effectiveness and versatility.

-  The figures presented in the paper are clear and informative.

**Weaknesses:**

- A key issue that remains insufficiently addressed throughout the paper is a clear justification for the insufficiency of traditional methods.

    For example, the noise present in the images (as illustrated in Fig. 4) appears to be predominantly thermal noise, a scenario where matched filter techniques are typically effective. In astronomical contexts, considerable effort is devoted to calibrating and characterizing both the noise properties and the point spread function (PSF) of the measurements. This careful calibration often enables traditional methods to perform well. In contrast, the application of deep learning for object detection is generally motivated in domains such as natural image analysis, where object representations are less well-defined and traditional approaches may fall short.

    Therefore, the paper would benefit from a more thorough discussion of the specific limitations of existing methods in this context, and a clearer justification for the adoption of deep learning techniques.

- Equation 7 appears to be somewhat counterintuitive. The transformation it defines takes the opposite of certain values within a specific range, which results in a modification of the overall ordering of the confidence scores. Since the c values are constrained to the interval [0,1] the ordering of any values within [0,\tau_{low}] is effectively reversed by this operation. It is not immediately clear what the intended benefit of this reversal is, as simply setting these values to zero would seem to achieve a similar effect in terms of suppressing low-confidence detections, without altering the relative ordering. Further clarification or justification for this particular transformation would be helpful.

- It would be helpful for the authors to clarify the criteria used by astronomers to differentiate between actual sources that require annotation and background blurry spots. From Fig. 4, it appears that many spots in the image are not considered stellar sources. This raises several important questions regarding the learning objective: What specific characteristics distinguish a true positive (i.e., a real astronomical source) from a false positive (e.g., noise or background artifacts)? Is the distinction primarily based on the shape of the spot, such as the degree of peakedness, with real stellar sources exhibiting a more pronounced peak? Are there additional features or contextual information that are taken into account during annotation? Furthermore, what are the typical origins of the spots that are not labeled as sources: are they due to instrumental noise, cosmic rays, or other astrophysical phenomena? In this context, how certain are the astronomers of the annotations ? A more detailed discussion of these criteria would provide valuable context for understanding both the annotation process and the challenges faced by automated detection methods.

- An important practical consideration for astronomers that appears to be missing from the proposed method is the quantification and control of uncertainty.  Does the method provide any mechanism for estimating or controlling the false alarm rate, for example through score calibration or threshold adjustment?

**Questions:**

- Were all the baselines in Table 1  trained on LAMOST ? Same question for Table 4 with the GED dataset ?

- L 263: “negative samples may contain important information that distinguishes the astronomical sources” : How so ? Can you elaborate on this ?


Minor:

- L 267: The term "matrix" is confusing in this context, maybe "confidence map" or "detection map" would be more appropriate.

- L 228:  “Fronzen” -> “Frozen”

---

### Official Review · Reviewer_ZfNx · 2025-11-05

**Soundness:** 2
**Presentation:** 3
**Contribution:** 2
**Rating:** 2
**Confidence:** 5

**Summary:**

This article deals with the detection of astronomical sources in a specific body of data coming from a spectroscopic telescope located in China. Sources can be both stars from our own galaxy or distant galaxies, in similar proportions. The authors propose a semi-supervised source detection method comprising several novel modules. The first is a source enhancement module that learns to enhance the contrast of the sources. The second is a pseudo-labelling technique based on confidence weighting. The third is a "cross-view" pseudo-label augmentation technique using two teachers.

The authors compare their source detection technique on other semi-supervised and supervised detectors not specifically tuned to astronomical source detection, and obtain better results. They perform an ablation study showing all three modules contribute to these results, and briefly test their method on a computer-vision benchmark, performing ellipse detection on natural-images, with state-of-the-art results.

**Strengths:**

The paper is reasonably well written and illustrated. The three contributions to semi-supervised learning are interesting although the first one (source-light enhancement module) is specific to astronomical source detection. The confidence-based module and the cross-view module work well together. The paper is sufficiently detailed, and with source code and data availability, should be reproducible. The approach does not require much computation power (a single inexpensive GPU used for training). Testing the approach on an ellipse-detection problem in computer vision show that some of the approaches carry over to different problems.

The comparison is well conducted on similar, competing methods, achieving state-of-the-art methods. The ablation study show that the proposed enhancement approaches for SSD work well together.

**Weaknesses:**

The paper focuses on solving a specific problem, namely source detection for a specific spectroscopic telescope. This is not uninteresting, and the authors do show that some of their approach can carry over to other problems -- albeit with less impressive success, but the fact remains that the vast majority of the paper focuses on this problem. By itself this is not bad, but it does limit the impact of the paper.

The paper would have been much more interesting if the authors had not focused on the data coming from the LAMOST telescope, but had tried to propose a more general astronomical source detector. As it stands, this telescope has a unique optical design and purpose and therefore uniquely specific data; whereas source detection in astronomy is a widely studied topic with many aspects (stars, galaxies, faint objects, asteroids etc) and many evolving challenges. The data samples shown in supplementary material seems to show exclusively stellar-like point sources marred by various, very visible, optical artefacts, that are not representative of the variety of astronomical sources found in widespread data releases such as the Sloan Digital Sky Survey found at https://www.sdss.org

It is to be noted that many astronomical databases have been publicly available for quite some time and some are already well annotated. See for example the popular Galaxy Zoo project (https://data.galaxyzoo.org) which crowdsourced the detection and classification of millions of deep sky objects. Therefore I'm skeptical of the widespread appeal of the author's data contribution.

Another point I find problematic is the authors' claim that only relatively recent object detectors based on deep learning have any merit. The author's comparison section only lists such methods. In astronomy, the standard source detector is the tool sextractor, which the authors do cite, which was introduced in 1996, but of course has been regularly updated. The latest version dates from 2017. The python version the author used dates from 2024. This is one of the most widely used software tool in all of astronomy. It works very well across a wide variety of astronomical imaging modalities. In fact the authors of this paper used this tool to make their annotations! It is unthinkable that they did not compare their performance to this tool. Note that various enhancements have been proposed to sextractor, see [1,2] among others. Also, due to its versatile but simple approach, sextractor results are highly interpretable, which is not the case for most deep-learning based detectors.

The authors claim "traditional" tools like sextractor based on statistical detection and classical hand-crafted features are no longer relevant since deep-learning based detection are both faster and better performing. In fact sextractor is very fast and runs on modest hardware, and the authors did not show in any way that they did get better results than sextractor.

Concerning the contributions, the first (source enhancement) is interesting but very specific. The second has larger appeal, although confidence-based pseudo annotations have been proposed before and have been shown to be biased. The third module is not very well described, in a single paragraph.


[1] Thanh Xuan Nguyen et al. CGO: Multiband astronomical source detection with component-graphs. In 2020 IEEE International Conference on Image Processing (ICIP), pages 16–20. IEEE, 2020.
[2] Mohammad Hashem Faezi, Reynier Peletier, and Michael HF Wilkinson. Multi-spectral source-segmentation using semantically-informed max-trees. IEEE Access, 12:72288–72302, 2024.

**Questions:**

- Why not compare with sextractor ?
- Why ignore more recent advances in astronomical source detection using traditional tools?
- The bibliography on source detection with deep learning is incomplete. See for example [3] for a list of recent methods.
- How much retraining would be necessary for the proposed method to work on say, SDSS data?
- How were the annotations produced exactly? How much interaction with astronomers was necessary? how long was the process?



[3] Ma Long, Xin Jiarong, Du Jiangbin, Zhao Jiayao, Wang Xiaotian, and Zhang Yu. Astronomical pointlike
source detection via deep feature matching. The Astrophysical Journal Supplement Series, 276(1):4, 2024.

---

### Note · Authors · 2025-11-14

I have read and agree with the venue's withdrawal policy on behalf of myself and my co-authors.